# Ancient proteins from ceramic vessels at Çatalhöyük West reveal the hidden cuisine of early farmers

Jessica Hendy [1,2], Andre C. Colonese[2], Ingmar Franz[3], Ricardo Fernandes[1,4], Roman Fischer [5], David Orton [2], Alexandre Lucquin [2], Luke Spindler[2,6], Jana Anvari[7], Elizabeth Stroud[4], Peter F. Biehl[8], Camilla Speller [2,9], Nicole Boivin[1], Meaghan Mackie [10,11], Rosa R. Jersie-Christensen[11], Jesper V. Olsen [11], Matthew J. Collins [2,10], Oliver E. Craig [2] & Eva Rosenstock [7]

The analysis of lipids (fats, oils and waxes) absorbed within archaeological pottery has revolutionized the study of past diets and culinary practices. However, this technique can lack taxonomic and tissue specificity and is often unable to disentangle signatures resulting from the mixing of different food products. Here, we extract ancient proteins from ceramic vessels from the West Mound of the key early farming site of Çatalhöyük in Anatolia, revealing that this community processed mixes of cereals, pulses, dairy and meat products, and that particular vessels may have been reserved for specialized foods (e.g., cow milk and milk whey). Moreover, we demonstrate that dietary proteins can persist on archaeological artefacts for at least 8000 years, and that this approach can reveal past culinary practices with more taxonomic and tissue-specific clarity than has been possible with previous biomolecular techniques.

[1] Department of Archaeology, Max Planck Institute for the Science of Human History, 07745 Jena, Germany. [2] BioArCh, Department of Archaeology, University of York, York YO10 5DD, UK. [3] Institute of Prehistoric and Protohistoric Archaeology, Christian-Albrechts-Universität zu Kiel, D-24098 Kiel, Germany. [4] School of Archaeology, University of Oxford, Oxford OX1 2PG, UK. [5] Target Discovery Institute, University of Oxford, Oxford OX3 7FZ, UK. [6] Oxford Radiocarbon Accelerator Unit, University of Oxford, 1 South Parks Road, Oxford OX1 3TG, UK. [7] Institute of Prehistoric Archaeology, Freie Universität Berlin, 14195 Berlin, Germany. [8] Department of Anthropology, University at Buffalo, Buffalo, NY 14261–0026, USA. [9] Department of Anthropology, The University of British Columbia, Vancouver, BC V6T 1Z1, Canada. [10] EvoGenomics, Natural History Museum of Denmark, University of Copenhagen, 2100 Copenhagen, Denmark. [11] Novo Nordisk Foundation Center for Protein Research, Faculty of Health and Medical Sciences, University of Copenhagen, 2200 Copenhagen, Denmark. Correspondence and requests for materials should be addressed to J.H. (email: hendy@shh.mpg.de) or to O.E.C. (email: oliver.craig@york.ac.uk) or to E.R. (email: e.rosenstock@fu-berlin.de)

The molecular analysis of archaeological artefacts has produced remarkable insights into culinary practices over the last 10,000 years. The analysis of lipids (fats, oils and waxes) associated with ancient pottery has been at the forefront of this field and fundamental for identifying the nature of prehistoric economies and consumption practices[1–4], particularly for dairy (milk, cheese) and insect (honey, beeswax) products, that are invisible to other methods of enquiry. As such, residue analysis is now routinely used in archaeological research and has been applied globally, including to some of the world's earliest pottery vessels[5]. In Southwestern Asia, Africa and Europe, lipid residue analysis has fundamentally advanced our understanding of the development of early pastoral economies. Together with ancient DNA analysis, this has shown that early, and most likely lactose intolerant, farmers consumed dairy products by the time that the appearance of pottery first allows for the detection of this foodstuff[1,6–8]. Although this research has far-reaching consequences for understanding the emergence of dairying and the subsequent evolution of cultural, dietary and economic practices, lipids often lack taxonomic and tissue specificity (i.e., which species and/or which part of the animal/plant was consumed). In addition, the resolution of lipid analysis can be diminished by the mixing of different food products and the reuse of vessels[9]. Moreover, whilst fat-rich animal products are often readily identifiable[10], there have been only a few reports of the identification of plant lipids in prehistoric pottery[11–13], potentially skewing our interpretation of past resource use and culinary practice. As a consequence, our narratives regarding the development and use of early pottery in Southwest Asia and Europe have been tightly bound to wider debates concerning domestication, animal management and secondary product exploitation.

The analysis of ancient proteins offers an alternative approach for identifying foodstuffs prepared in ceramics[14,15], providing the opportunity for improved tissue and taxonomic resolution. Compared to lipids, proteins are at much greater concentration in plant foods, such as cereals and legumes, thus also providing scope for identifying a greater range of products. Previously, immunological detection of proteins has been attempted for ceramic artefacts[14,16,17]; however, this approach is confined to detecting pre-selected proteins of interest and requires the survival of specific target epitopes (regions of a protein which enable antigen–antibody binding). In the last decade, immunological methods have been superseded by liquid chromatography-tandem mass spectrometry (LC-MS/MS), which has been increasingly applied to archaeological materials[18]. Such 'shotgun' proteomic approaches are able to reveal a wide range of proteins in a sample, can detect fragmented and denatured proteins, and are consequently less impacted by protein degradation. This approach has been successfully used to investigate food and other proteins trapped in calcified dental plaque, also known as dental calculus[19,20]. However, the application of shotgun proteomics to archaeological artefacts has so far been limited to exceptionally well-preserved examples from waterlogged[21], cold[11,22] or arid contexts[23]. Beyond such exceptional contexts, it has not been clear whether dietary proteins would survive following long-term exposure to the burial environment[24], whether sampling and extraction strategies could be made more effective at recovering food proteins from vessel walls[25], or whether dietary proteins would survive food preparation, particularly protracted heating.

Here we apply a shotgun proteomic approach to ceramic sherds from the early farming site of Çatalhöyük (ca. 7100–5600 cal BC[26–28]) in central Anatolia. Specifically, sherds derive from the West Mound (Fig. 1), radiocarbon dated to 6000–5600 cal BC[28], a stage within a process of socio-economic change (beginning on the East Mound at ca. 6500 cal BC), including a shift from community to household-centred economies, changing patterns of landscape use and a greater diversification in pottery use[29–34]. These processes also roughly coincide with similar developments in Upper Mesopotamia and the Levant, as part of what has been termed the Second Neolithic Revolution[35–37], and the start of the spread of farming into Western Anatolia[38–40]. Previous organic residue analysis undertaken at Çatalhöyük focused on identifying animal fats from a range of vessels from the earlier phases (6800–6300 cal BC) at the East Mound in order to investigate early animal management strategies[1,41]. Here we analyze vessel sherds from the later occupation on the West Mound of the site, stemming from building infills in Trench 5 (Supplementary Figure 1) dating to a narrow time interval between 5900 and 5800 BC[28]. The vessel sherds (Fig. 2a) can be reconstructed as open bowls and jugs (Fig. 2a), and show minimal signs of post-firing exposure to heat with no extensive exterior soot marks. Thin (<2 mm) calcified deposits are present on the interior surfaces and appear to have formed during use (Fig. 2b). These calcifications do not appear to be post-depositional limescale build-up, as we find little depositional build-up on the outer wall, nor on the edges of the broken sherds (Fig. 2d). Moreover, no depositional build-up was recorded on bone fragments excavated from the same contexts. Similar limescale deposits are readily formed on modern cooking pots in this region (Fig. 2c). X-ray diffraction (XRD) and X-ray fluorescence (XRF) analyses on the archaeological sherd deposits reveal that these deposits consist of calcium carbonate ($CaCO_3$). We apply a protein extraction methodology based on Gel-Aided Sample Preparation (GASP) to these deposits, as well as to samples of the inner and outer ceramic wall, and complement this with isotopic and molecular identification of extracted lipids. These data reveal mixes of cereals, pulses, dairy and meat products, and

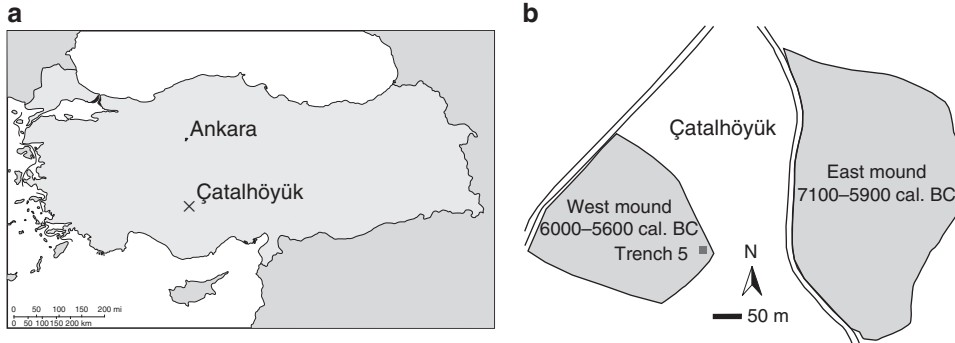

**Fig. 1** Map of Çatalhöyük. **a** Site location and **b** site plan of Çatalhöyük, adapted from Hodder[33]. Çatalhöyük consists of two distinct mounds; the East Mound, dating to ca. 7100–5900 cal BC[26, 27], and the West Mound, dating to ca. 6000–5600 cal BC[28]

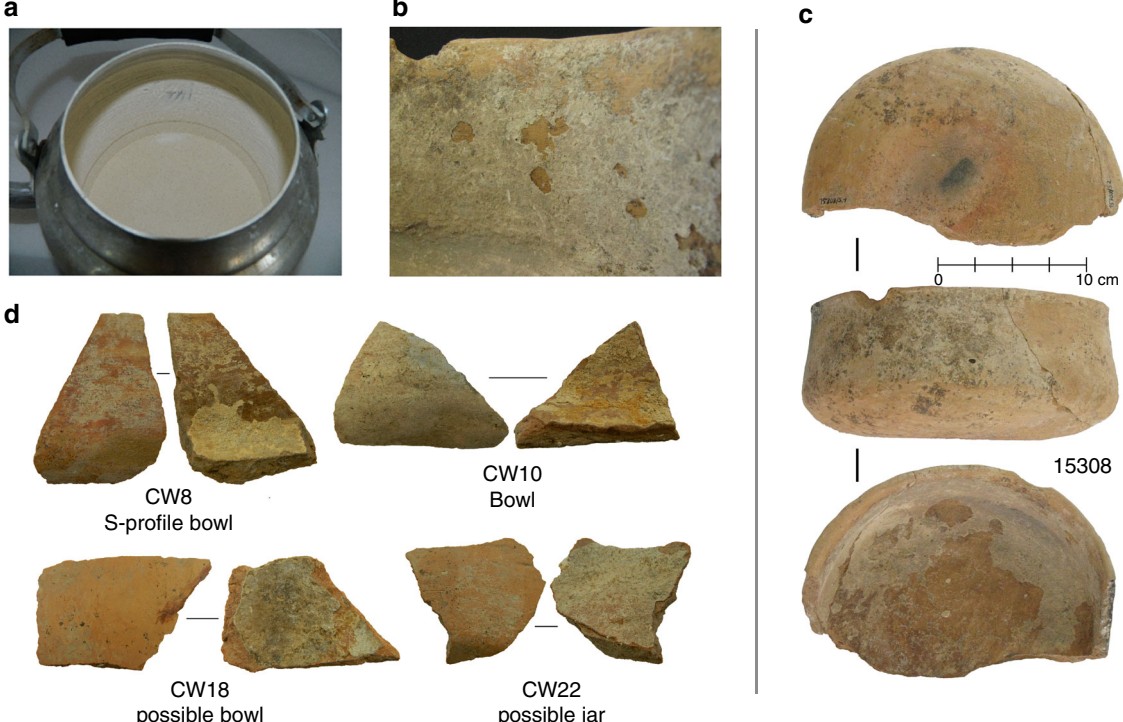

**Fig. 2** Examples of calcified deposits from modern and ancient vessels at Çatalhöyük. **a** Examples of CaCO$_3$ accretions from a modern tea water pot with extensive calcified deposits used near the research project compound Çatalhöyük, **b** a close-up of calcified deposits, **c** a relatively intact vessel (not analyzed in this study) demonstrating bowl shape and extent of calcified deposits and **d** a selection of four sherds analyzed in this study showing deposits adhering to the inside surface of the ceramic sherds

demonstrate the survival of food proteins on ceramic residues for up to ca. 8000 years.

## Results

**Evidence of dairy, cereals, legumes and non-dairy animal proteins**. The sequence analysis of tryptic peptides revealed a range of dietary foodstuffs, including dairy (Bovinae, Caprinae, *Ovis*, *Capra hircus*), cereal grains (*Hordeum vulgare*, *Triticum*, Triticeae), legumes (*Pisum sativum* and *Vicia* sp) and non-dairy animal proteins (Caprinae, Cervidae and Bovinae haemoglobin) (Fig. 3; Supplementary Data 2). These products are consistent with botanical and faunal records from the site and region[38], but, for many of these foodstuffs, these findings represent their first secure in situ species identification on pottery. A greater number of identified proteins was found in the calcified deposits compared to the ceramic walls (Fig. 3), suggesting that these limescale deposits may provide a stable environment for protein preservation, an observation that has important implications for pottery conservation. Of the 10 calcified deposits analyzed, all but two (CW10 and CW24) yielded proteins from foodstuffs; more than one foodstuff was detected in five vessels, indicating the mixing of products in culinary practices and/or sequential use. Two morphologically distinct bowl types were analyzed, with no consistent differences in foodstuff content observed between them, in contrast to the only jar analyzed (CW22).

Calcified deposits and ceramic samples yielded proteomic evidence of dairy products in the majority of vessels and in the single jar analyzed. Milk proteins were predominantly derived from Caprinae (sheep and goats), and several peptides could be assigned more specifically to *Ovis* (sheep) and *Capra* (goat) (Fig. 4). One vessel (CW11) yielded two milk proteins specific to Bovinae with no other species matches to milk, which may suggest a more dominant use of cow's milk in this vessel. Whilst protein preservation in the calcified deposits was exceptional, milk proteins were also detected in three of the five samples of the ceramic walls, showing this approach may also be applicable to potsherds without calcified deposits (Fig. 3). Haemoglobin proteins, which are predominantly found in blood plasma and animal tissues, were also detected in calcified deposits and ceramic matrix samples and were primarily derived from Caprinae, and to a lesser extent Cervidae and Bovinae. Other muscle and meat-derived proteins such as myosin and titin were not detected, thus the incorporation of haemoglobin into calcified deposits and the penetration into the ceramic matrix may be related to the solubility of this globulin protein. Interestingly, haemoglobin from Caprinae and *Equus* was detected on the outer vessel wall of two ceramic samples (CW10 and CW20), both of which had red paint on the outer wall (Supplementary Data 1), and the whey protein beta-lactoglobulin was detected on the outer vessel wall of CW22, which may suggest the use of animal products in post-firing treatments or the pollution of the outer wall as a result of pouring or spilling. The identification of ruminant-derived animal proteins is consistent with zooarchaeological remains identified at Çatalhöyük West, which are dominated by Caprinae (>90%), followed by *Bos*—believed to be predominantly domestic—and wild equids[42]. To identify whether these Caprinae remains derive from sheep or goat, ZooMS collagen fingerprinting (zooarchaeology by mass spectrometry) was performed (Supplementary Data 4). These results suggests a domestic *Ovis:Capra* ratio of approximately 6:1, a dataset consistent with the species proportions observed through protein evidence, i.e., a dominance of Caprinae, primarily *Ovis*, over Bovinae.

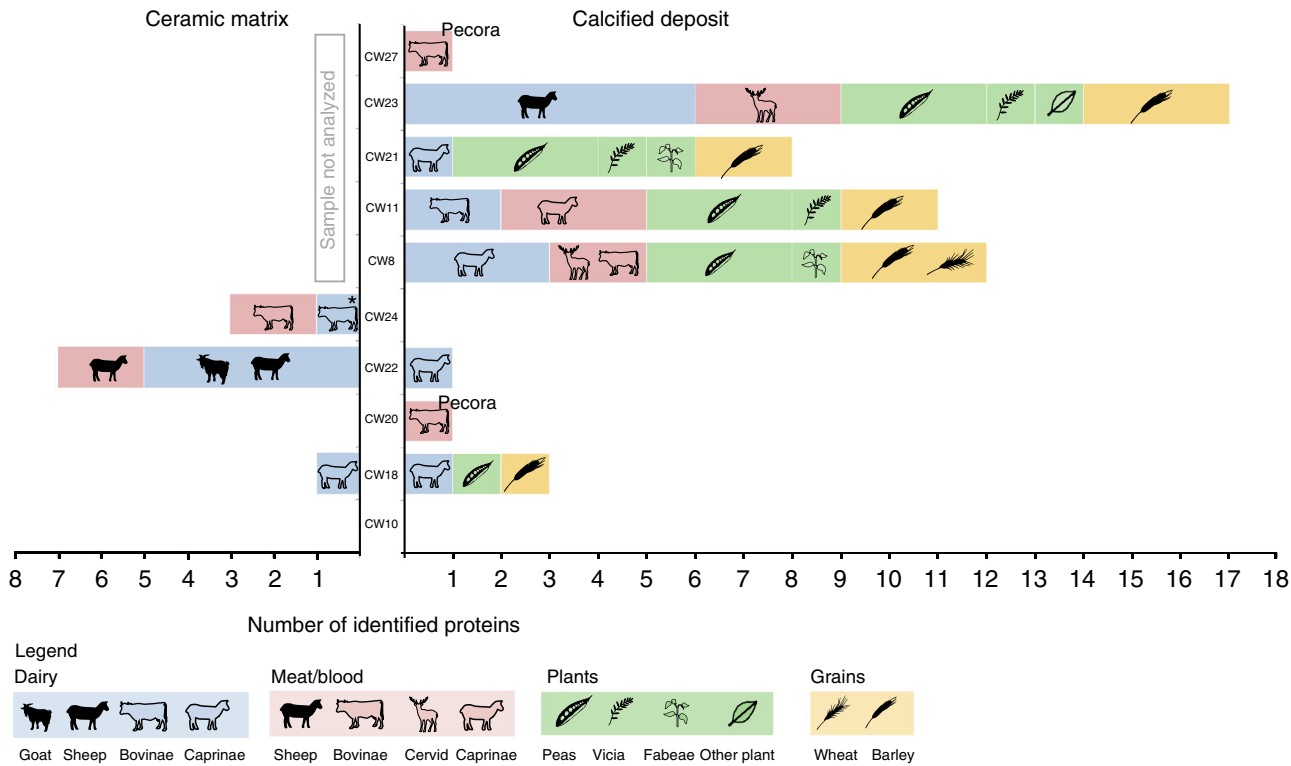

**Fig. 3** Summary of dietary-derived protein identifications. The left graph summarizes proteins extracted from the ceramic matrix of the sherd's interior wall and the right graph summarizes proteins extracted from calcified deposits adhering to the inner wall. Filled icons represent protein taxonomic assignments to the genus or species level, while transparent icons represent identifications to higher taxonomies (subfamily, family). In CW20 and CW27, haemoglobin was identified to the taxonomic level of Pecora. In CW24, the milk protein beta-lactoglobulin could be assigned to either Bovinae or *Ovis*

**Evidence of ruminant dairy and ruminant adipose fats**. To complement protein analysis, lipids were extracted from the same sherds and analyzed by gas chromatography-mass spectrometry (GC-MS) (Supplementary Table 2, Supplementary Table 4). Saturated and unsaturated fatty acids, dihydroxy fatty acids, dicarboxylic acids and *n*-alkanes were released from both the calcified deposits and ceramic samples using an acid/methanol-based extraction. These compounds could potentially be derived from a wide range of animal and plant products[43]. The presence and diastereomeric ratio of phytanic acid and the presence of branched chain $C_{15}$ and $C_{17}$ fatty acids found in many of the sherds is indicative of ruminant products[10,44] whereas the presence of *n*-alkanes with odd number of carbon atoms ($C_{25}$ to $C_{35}$) may be derived from degraded plant waxes[13]. Calcified and ceramic samples were also extracted using established methods to determine the presence of acyl lipids (mono-, di- and triglycerides)[45,46], but extensive degradation has prevented their survival in the samples. To investigate further, the carbon isotope ratios of the main fatty acids ($C_{16:0}$ and $C_{18:0}$) were determined by gas chromatography-combustion-isotope ratio mass spectrometry (GC-C-IRMS). This approach is commonly used to distinguish ruminant dairy, ruminant adipose and non-ruminant fats based on the difference in the carbon isotope values between these fatty acids ($\Delta^{13}C = \delta^{13}C_{18:0} - \delta^{13}C_{16:0}$)[47].

Our results from the calcfied deposits from the West Mound have a similar range of $\Delta^{13}C$ values to previously analyzed pottery from the Çatalhöyük East Mound[1,41], and point to a dominance of ruminant adipose tissues (Fig. 5a). In contrast to the protein results, only one sample, a jar (CW22), produced a $\Delta^{13}C$ value consistent with the published reference values for dairy products. Interestingly, for the remaining vessels, all of which are bowls, fatty acids from the calcified deposits were generally more

enriched in $^{13}C$ compared to those absorbed within the vessel walls. Three of the ceramic matrix extracts had stable carbon isotope values matching reference plant values (Fig. 5a) but also conceivably other non-ruminant fats, including reference wild boar from the Middle East[48] and wild equids. The discrepancy between the calcified deposits and the ceramic matrix may be due to differential binding of fats in these two matrix types or else mixing of different proportions of isotopically distinct foodstuffs during residue formation, as is suggested by the protein results.

**Mixing model**. The Çatalhöyük vessels have a wider range of $\delta^{13}C$ values compared to European prehistoric sites and encompass values for domestic animals grazing on $C_4$ plants[46]. This is supported by isotope analysis of animal bone collagen previously undertaken at this site[49]. It is often asserted that $\Delta^{13}C$ values are insensitive to animal diet but rather respect physiological differences between sources and therefore are globally applicable[1]. Whilst this may be true, previous studies have not considered the effects of mixing animal fats with plant oils. Plants have a much higher $C_{16:0}$ to $C_{18:0}$ ratio than animal products, and may produce significant deviations in the $\Delta^{13}C$ values depending on the absolute $\delta^{13}C$ of the end-members. To investigate further, using a mixing model, we examined how $\Delta^{13}C$ values respond to a range of simulated scenarios where dairy products from animals with variable $C_3$ and $C_4$ diets are mixed with $C_3$ plants, e.g., legumes and wheat/barley. We show that $\Delta^{13}C$ values consistent with ruminant adipose fats are probable when even modest amounts of plant lipid (ca. 20%; Fig. 5b) are mixed with dairy. For example, after accounting for the different lipid contents of the foodstuffs, a ruminant adipose value could be obtained by mixing ca. 750 g of barley (2.5 wt% lipid) with 1 L of raw sheeps' milk (7 wt% lipid). In contrast, it was impossible to produce $\Delta^{13}C$ values within the

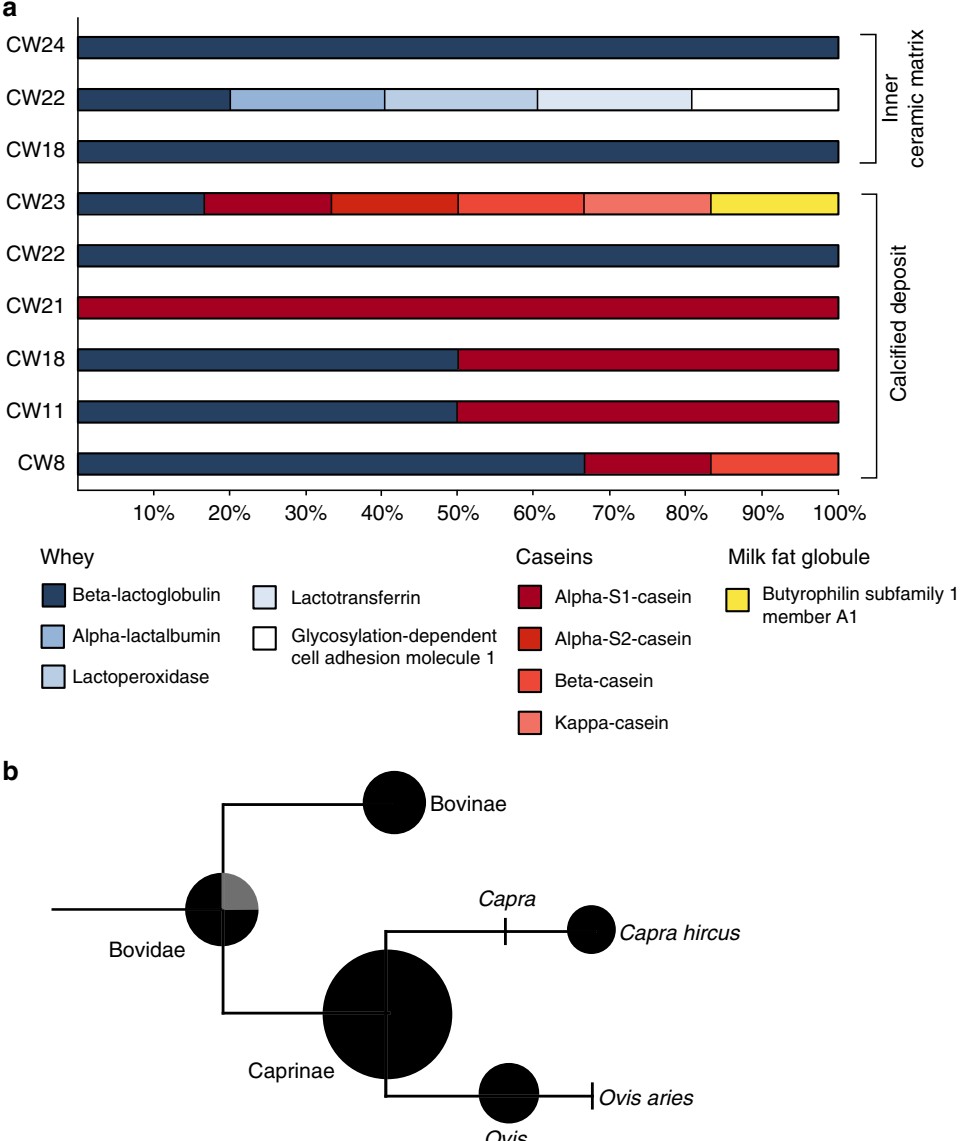

**Fig. 4** Proteomic evidence of dairy products in vessels from Çatalhöyük West Mound. **a** The proportion of identified caseins (red), whey proteins (blue) and milk fat globule-associated proteins (yellow) in calcified deposits and inner-wall ceramic samples, **b** a schematic cladogram indicating the taxa of milk proteins extracted from samples of calcite and the inner ceramic vessel walls. Counts of identified dairy proteins are represented as scaled pie charts. For the Bovidae pie chart, black represents proteins assigned to Bovidae, while grey indicates those which could not be unambiguously assigned below Bovinae/Ovis. This is due to the presence of a genus-specific polymorphic amino acid in the identified peptide TPEVD(D/N/K)EALEK, where D is specific to Bovinae, N is specific to Ovis and K is specific to Capra. Because N deamidates to D, the presence of a D at this position cannot be unambiguously assigned to Bovinae or Ovis

ruminant dairy range when ruminant adipose fats (from animals with $C_3$ and $C_4$ diets) were hypothetically mixed with $C_3$ plants (Fig. 5c) meaning that identification of dairy foods using $\Delta^{13}C$ is robust in this scenario, although also we note that dairy $\Delta^{13}C$ values are easily generated by mixing $C_3$ ruminant adipose fats with $C_4$ plants.

These simulated models highlight the insensitivity of the lipid isotope approach to mixtures; dairy fats cannot be ruled out from contributing to the ceramic matrix and plant lipids might have been combined with dairy in five of the calcified deposits analyzed (Fig. 3, Supplementary Figure 3). In light of the protein results, we suggest that such food mixtures are entirely plausible. Yet, without further knowledge of the isotopic end-members for the different foods, which may include $C_3$- and $C_4$-fed ruminant and non-ruminant animals and a wider range of plants than currently

measured, further interpretation or more accurate quantification using stable isotope analysis cannot be made.

## Discussion

Our study presents a more nuanced picture of early farming pottery use compared to previous ceramic studies that have been methodologically constrained to the identification of lipid-rich animal products. The protein evidence indicates that cereals and legumes were processed in pottery from Çatalhöyük West, but these foodstuffs were undetectable using GC-MS- and GC-C-IRMS-based approaches. It is conceivable that other pottery vessels from early farming sites in Anatolia, the Near East, Europe and Africa, also had a much wider range of uses, beyond the processing of milk and meat as often suggested by their

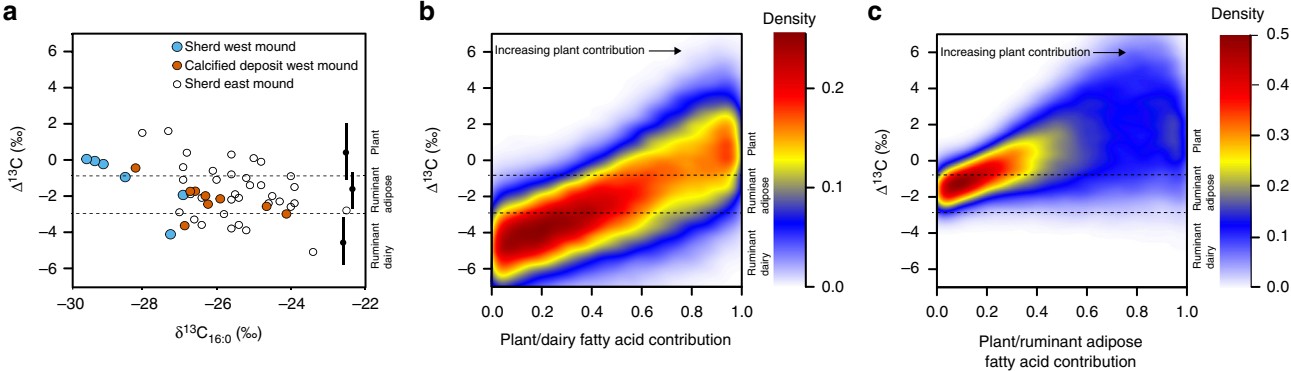

**Fig. 5** Lipid stable isotope characteristics from archaeological samples against theoretical mixes. **a** Plot of $\Delta^{13}C$ against $\delta^{13}C_{16:0}$ compared with reference ranges for authentic plant, ruminant adipose and dairy fats (mean ± 1σ). Published isotope data from the East Mound (ca. 6800–6300 cal BC) of Çatalhöyük[41] are displayed for comparison. Density distributions of $\Delta^{13}C$ values obtained by theoretical mixing of **b** dairy and **c** ruminant adipose fats with an increasing amount of $C_3$ plant lipids

predominance in lipid analysis[1,46,47,50]. If so, the invention of pottery in the Near East at the end of the 8th millennium BC[51] and its subsequent diffusion with the expansion of the Neolithic after the mid 7th millennium[52], which at Çatalhöyük corresponds to the intensification of agricultural practices[33,53], may have also been driven by the need to process agricultural produce rather than simply animal products alone.

The detection of dairy proteins is consistent with previous lipid research[1,54] showing that this practice dates to at least the 7th and 6th millennia cal BC in the Near East and Europe, respectively. This follows initial animal domestication more rapidly than previously thought[55,56] and most probably well before the establishment of lactose-tolerant adult populations[6]. For the first time, however, we can confirm through the identification of species-specific proteins that milking of three domesticated ruminants was practiced in Southwest Asia at broadly the same time. Moreover, protein analysis provides the first clues to the mode of early dairy production in prehistory. Peptides attributable to casein proteins were typically detected in calcified deposits, whereas milk proteins extracted from vessel walls were all derived from whey, particularly beta-lactoglobulin (Fig. 4). Such a signal could be anticipated if curd and whey separated in these vessels—a common practice for making many dairy products—as soluble whey proteins may more easily penetrate the ceramic matrix. Processing of fresh into soured milk may not only have helped preserve milk in the absence of cooling facilities, but was likely also important in rendering milk more digestible to early Eurasian farmers who were presumably lactose intolerant[6]. In the only jar tested (CW22), only whey proteins were detected in both ceramic walls and calcified deposits, a result more consistent with the handling of pure whey. This was the only vessel to produce a clear dairy lipid isotopic signal suggesting a dedicated use for this purpose.

Similarly, we can also use the tissue specificity of the proteins identified to comment further on plant processing. For the cereals, the proteins derived from barley (Hordeum vulgare) and wheat (Triticum sp.) are all expressed in the grain endosperm (Supplementary Data 2), indicating that this part of the plant was contained in these vessels, possibly as part of a porridge or soup. This fits with evidence from scanning electron microscope (SEM) studies of amorphous charred fragments of cereal products for an increase in porridge-like products in the later phases at Çatalhöyük (6400–6000 cal BC), after an initial dominance of bread type remains in earlier phases[57]. Comparing the proteomic results with the archaeobotanical record at Çatalhöyük West, we observed differences in the diversity and prevalence of particular

plant species. One notable absence in the proteomic data is lentils (Lens culinaris), which are frequently observed in the archaeobotanical record[58].

Archaeobotanical remains show a diversity of plant remains at Çatalhöyük West, such as cereals (einkorn wheat, emmer wheat, new-type glume wheat, free-threshing wheat and barley), wild mustard, pulses (lentil, pea, bitter vetch, chickpea, grass pea) and fruit and nuts (hackberry, pistachio and Prunus species)[58,59]. Macrobotanical evidence is based on the presence of charred botanical remains, which may be unrelated to their specific culinary context. In contrast, analysis of ceramic vessels may reveal the links between food processing and material artefacts, providing more direct evidence of cooking technologies and cuisine. However, shotgun proteomic approaches are heavily dependent on reference sequence databases, and many plant species are not represented or have limited representation, resulting in potential biases towards the detection of certain plant species. For example, there are significantly fewer reference protein sequences for bitter vetch (Vicia ervilia) than for wheat (Triticum aestivum), a well-studied domesticate (144,455 protein accessions–the whole proteome–in TreEMBL compared with six vetch proteins).

This study also highlights a new source of ancient biomolecules, in the form of calcified deposits lining the inside of archaeological vessels. Our data support the idea that mineral-organic binding facilitates long-term archaeological preservation of proteins and peptides, as shown with the recovery of proteins from other mineralized deposits, such as dental calculus[19,60], eggshell[61] and bone[62–64]. Such deposits are not unusual in archaeological contexts[65,66] and, although the chemical and mechanical removal of encrustations in ceramic vessels are still considered good practices in archaeology[67,68], our results highlight the rich biomolecular value of these in situ deposits, and we put out a call to retain them during post-excavation processing and cleaning.

## Methods

**Sample information.** Ceramic sherds derive from fills deposited within abandoned buildings unearthed in Trench 5 of the Çatalhöyük West Mound (Supplementary Figure 1). These finely layered deposits accumulated in the interiors of Trench 5 buildings during phases when they were not inhabited as dwellings and result from middening, craft production, informal storage, and also include the burial of two neonates. Stratigraphically, these episodes of middening/informal use are interlaced with episodes of habitation and construction, showing that Trench 5 buildings alternated in use. Although considered as a secondary refuse context, midden deposits are therefore part of the house formation and use history. Further, the stratigraphic association of the pottery with articulated animal bones attests to the rapid deposition of the finds assemblage[69]. This rapid deposition is supported by a

narrow time window of fills between ca. 5900 and 5800 cal BC according to modelling of the associated [14]C dates[28]. Samples for this project were chosen only from contexts in undisturbed stratigraphic sequences, preferably below an intact plaster floor or deep within the infill of a building.

Three vessel types were selected in this analysis; bowls ($n = 4$), s-profile bowls ($n = 5$) and one jar. These types represent almost the entire spectrum of vessel shapes at Çatalhöyük West known so far, and are thought to be associated with food consumption or presentation. The calcareous–ferruginous ceramic matrix belongs to the so-called light line fabrics consisting of a mixture of locally available raw materials, and vessels are often painted with reddish-brown pigment on the exterior and interior (Supplementary Data 1). The interiors of the sherds contained a solid, flaky material, with a layered appearance of white, cream and/or yellow-cream colour (Fig. 2). To test whether this material may be calcitic, a small subsample (<10 mg) of one of the deposits was removed and 2 M HCl was added. The bubbling produced from carbon dioxide production suggested that this deposit may indeed be calcitic in nature.

**Experiment summary**. We performed three sets of experiments (Supplementary Data 1, Supplementary Table 1); the extraction and LC-MS/MS analysis of ancient proteins from calcified deposits lining the inside of ceramic sherds ($n = 10$), ceramic matrix from the inner vessel wall ($n = 5$), a sample of burnt residue adhering to the inner vessel wall ($n = 1$) and from ceramic matrix of the outside of the vessel ($n = 3$) termed "Protein Analysis #1" in Supplementary Data 1 and Supplementary Table 1, a second protein extraction and complementary LC-MS/MS analysis on calcified deposits, inner and outer ceramic samples where sufficient material was available (16 of 19 samples) termed "Protein Analysis #2" in Supplementary Data 1 and Supplementary Table 1, and a lipid extraction and analysis on samples of calcified deposits ($n = 9$), ceramic matrix ($n = 5$) and of the outer vessel wall ($n = 3$). Tryptic peptides were extracted from demineralized calcite deposits and samples of the ceramic matrix using GASP protocol based on Fischer and Kessler[70] modified for archaeological samples. Lipids were extracted according to established methods[5,71].

**XRF and XRD analysis**. XRF and XRD analyses were performed on one sample to determine the composition of the accretion. XRF was carried out using a Horiba XGT-7000 X-ray Fluorescence Spectrometer equipped with a rhodium source. XRF analysis demonstrates the predominance of calcium with rough quantification of atoms heavier than sodium of 77% Ca, 21% P, 2% Fe. XRD was performed using a Bruker D8 powder X-ray diffractometer equipped with a copper source. XRD and XRF analysis indicates the presence of $CaCO_3$ in the form of calcite with the possible presence of traces of calcium phosphates/hydroxyphosphates.

**Protein extraction of calcified deposits and ceramic sherds**. Samples of deposit and ceramic matrix were removed from the sherds by scraping with a sterile dental scaler and a sharp, metal spatula and tryptic peptides extracted using a previously published GASP protocol[11,70]. To remove any potential surface contamination, 1 mL of EDTA (1 M) was added to the powdered deposit and rotated for 5 min, then centrifuged at 13,000 RPM for 5 min and the supernatant removed. A further 1 mL of EDTA was then added to the residual sample pellet and demineralized for 5–7 days. Samples were then spun at 13,000 RPM for 5 min and 950 μL of the supernatant removed; 5 μL of SDS (20%) and 45 μL of M-PER (Mammalian Protein Extraction Reagent, Thermo Fisher) was then added to the remaining 50 μL of supernatant and pellet and shaken for 15 min at room temperature, followed by an addition of 50 μL of DTT (1 M) and further shaking for 30 min at room temperature. 100 μL of Proto-Gel (37.5:1 Acrylamide to Bisacrylamide, National Diagnostics) was added and gently resuspended to mix, then left on the benchtop for 20 min; 8 μL of TEMED, followed by 8 μL ammonium persulfate (10% w/v solution), was added and gently mixed to polymerize the gel. The gel was then shredded into pieces to increase the surface area by passing the gel through a plastic grid inset by pulse centrifugation. The gel pieces were then fixed with methanol/water/acetic acid/ (50/40/10) solution and rotated for 30 min. The solution containing the gel pieces was then centrifuged and then the supernatant (the fixing solution) discarded. A series of washing and drying steps were then performed to exchange buffers. 1 mL of acetonitrile was added to dehydrate the gel pieces, rotated for 3 min, and the supernatant removed and discarded; 1 mL of urea (6 M) was added to the dehydrated gel pieces and rotated for 3 min. The urea solution was then removed with two acetonitrile drying steps: 1 mL of acetonitrile was added to partially dehydrate, rotated for 3 min, briefly centrifuged, then the supernatant removed. A further 1 mL of acetonitrile was added to fully dehydrate the gel pieces, rotated for 3 min and the supernatant removed. A further 1 mL of urea (6 M) was added and subsequently removed using acetonitrile dehydration. 1 mL of ammonium bicarbonate (0.05 M) was added to the dried gel pieces and rotated for 3 min. The ammonium bicarbonate solution was then removed with two acetonitrile drying steps: 1 mL of acetonitrile was added to partially dehydrate, rotated for 3 min, briefly centrifuged, then the supernatant removed. A further 1 mL of acetonitrile was added to fully dehydrate the gel pieces, rotated for 3 min and the supernatant removed. 200 μL of ammonium bicarbonate (0.05 M) and trypsin (5 μL of 0.5 μg/μL) was added to the dried gel pieces, rotated for 3 min to mix, and left to digest at 37 °C overnight.

Digested peptides were then extracted from the gel. First, the gel pieces were dehydrated with acetonitrile; 200 μL of acetonitrile was added to partially dehydrate the gel pieces, rotated for 5 min, pulse centrifuged, then the supernatant (containing peptides) transferred to a fresh tube. To extract acidic peptides, 200 μL of 5% formic acid solution was added to the gel pieces, rotated for 5 min and pulse centrifuged; 200 μL of acetonitrile was then added to the gel pieces, rotated for 5 min, pulse centrifuged and the supernatant (containing peptides) transferred to the tube containing the first fraction. To fully dehydrate the gel, 200 μL of acetonitrile was added, rotated for 5 min, pulse centrifuged, and the supernatant transferred to the tube containing the tryptic peptides. The extracts were then dried in a centrifugal evaporator and desalted using C18 Zip-Tips (Millipore) prior to MS/MS analysis.

**MS/MS analysis**. Tandem Mass Spectrometry was performed on an Orbitrap Fusion Lumos (Thermo Fisher) at the Mass Spectrometry Laboratories of the Target Discovery Institute at the University of Oxford. Dried peptides were resuspended in 20 μL of 0.1% trifluoroacetic acid and 2% acetonitrile before injection of 6 μL into the nLC-MS/MS. Chromatographic peptide separation was performed on a 50-cm easy spray column (Thermo Scientific) and a linear acetonitrile gradient from 2–35% in DMSO (5%) and formic acid (0.1%). Precursor peptides were detected with up to 50 ms accumulation time for an ion target of 4E5, followed MS/MS data acquisition for up to 3 s and a maximum parallel injection time of 250 ms per precursor mass. Precursors were isolated in the quadrupole with 1.2 Th and, following automatic exclusion for 60 s, selected at an intensity of 5000 or higher. MS1 spectra were acquired with a resolution of 120,000, while MS/MS spectra were acquired after CID fragmentation (35% collision energy) in the linear ion trap in rapid scan mode. Blank instrument washes were run between samples to monitor peptide carryover[61].

As a complementary line of evidence, a second batch of extractions was performed on the samples where available (16 of 18 samples) (Supplementary Table 1). These extracts were sent to the Novo Nordisk Centre for Proteomics in Copenhagen, and analyzed on a Q-Exactive HF Orbitrap tandem mass spectrometer (Thermo Scientific). The dried peptides were resuspended in 10 μL 0.1% trifluoroacetic acid and 5% acetonitrile, sonicated, transferred to 96-well plate, and 5 μL injected using an EASY-nLC 1000 system (Thermo Scientific). The MS protocol was then followed as described in Demarchi et al.[52]. The samples were separated with a 165 min gradient of 2–60% 0.1% TFA, using a 50-cm PicoFrit column (75 μm inner diameter) in-house packed with 1.9-μm C18 beads (Reprosil-AQ Pur, Dr. Maisch). The Q-Exactive HF was operated in data-dependent top 10 mode with full-scan mass spectra recorded at a resolution of 120,000 with a target value of 3e6 and a maximum injection time of 20 ms. HCD-fragmented ions were recorded at a resolution of 60,000 with a maximum ion injection time set to 108 ms and a target value set to 2e5. Blank instrument washes were run between samples to monitor peptide carryover[61].

**MS/MS data analysis**. Raw spectral data were converted to Mascot generic format (mgf) using Proteowizard MSConvert (version 3.0.4743) using the 100 most intense peaks. MS/MS ion database searching was performed on Mascot (Matrix Science[TM], version 2.4.01) against UniProtKB (17–03–2016 version). Searches were made against a decoy database to estimate false discovery rates (FDR), which were adjusted to 5% PSM homology. The following modifications were set during database searching; propionamide (C) was set as a fixed modification, and acetylation (protein N-terminus), deamidation (NQ), methionine oxidation, propionamide (K) and propionamide (N-terminus) were set as variable modifications. Peptide tolerance was 10 ppm, and MS/MS ion tolerance was 0.5 Da (for the Fusion Lumos samples) and 0.07 Da (for the Q-Exactive HF samples). Fully tryptic peptides were searched with up to one missed cleavage. Results were adjusted to a 5% FDR (PSM homology) and having a minimum of two peptide matches. Subsequently, peptides of plant and non-human animal origin were interrogated using BLAST (NCBI), where peptides were aligned using BLASTp against all non-redundant nucleotide sequences to determine their taxonomic specificity (Supplementary Data 2, "Peptide Identifications"). Protein expression data were retrieved, where available, from UniProtKB. To compile the final list of identified dietary proteins (Supplementary Data 2; Fig. 3), identified peptides from both sets of LC-MS/MS experiments were pooled and any duplicates deriving from the two sets of experiments identified as one identification (Supplementary Data 2, "Protein Summary").

**Lipid extraction**. Lipids were directly extracted and methylated with acidified methanol according to established methods[5,71] to yield an acid/methanol extract (AE). Briefly, methanol was added to homogenized ceramic powders (0.5–1.2 g) or calcified deposits (70–360 mg) (Supplementary Table 2). Blank extractions were included with each batch. The samples were sonicated in a water bath for 15 min, and acidified with concentrated sulphuric acid (200 or 800 μL). The acidified suspension was heated in a block for 4 h at 70 ° and extracted with $n$-hexane ($3 \times 2$ mL) for subsequent analysis. Where available, a separate portion of each sample (ceramic powders; 0.5–1.2 g, calcified deposits; 70–360 mg) was solvent extracted (DCM:MeOH; 2:1 v/v, $3 \times 2$ mL, 15 min) using established protocols to prepare a total lipid extract (TLE) with the aim of identifying acyl

lipids (mono-, di- and triglycerides), which are hydrolyzed using the acid/methanol protocol. These samples were silylated before analysis by HT-GC-MS-FID.

**GC-MS.** GC-MS was performed on acid/methanol extracts using a 7890A Series chromatograph coupled to a 5975C Inert XL mass-selective detector (MSD) with a quadrupole mass analyzer (Agilent Technologies, Cheadle, UK). Helium was the carrier gas used with a constant inlet/column head-pressure. A splitless injector was used and maintained at 300 °C and the GC column was inserted directly into the ion source of the mass spectrometer. Spectra were obtained by scanning between 50 and 800 $m/z$ with the ionization energy of the mass spectrometer at 70 eV. Two different column phases were used. General screening was performed using a DB-5ms (5%-phenyl)-methylpolysiloxane column (30 m × 0.250 mm × 0.25 μm; J&W Scientific, Folsom, CA, USA). The temperature for this column was set at 50 °C for 2 min, then raised by 10 °C/min to 325 °C, where it was held for 15 min. TLEs were analyzed with a HT-DB1 GC-MS-FID, 100% dimethylpolysiloxane (15 m × 0.320 mm × 0.1 μm) (J&W Scientific, Folsom, CA, USA) column. The injector was maintained at 350 °C. The temperature of the oven was set at 50 °C for 2 min, and then raised by 10 °C/min to 350 °C, where it was held for 15 min. The column flow was split 9:1 (MSD: Flame ionization detector), with the MSD conditions described above.

**Gas chromatography-combustion mass spectrometry.** GC-C-IRMS analysis was carried out using an Isoprime 100 (Isoprime, Cheadle, UK) coupled to a Hewlett Packard 7890B series GC (Agilent Technologies, Santa Clara, CA, USA) with an Isoprime GC5 interface (Isoprime Cheadle, UK). One microlitre of each sample was injected into DB-5MS ultra-inert fused-silica column. Eluted products were ionized in the mass spectrometer by electron impact and ion intensities of 44, 45 and 46 $m/z$ were recorded for automatic computing of the $^{13}C/^{12}C$ ratio of each peak. Computation was made with IonVantage and IonOS softwares (Isoprime, Cheadle, UK) based on comparisons with standard reference gas ($CO_2$) of known isotopic composition that was repeatedly measured. Results were expressed in per million (‰) relative to an international standard, V-PDB, and the accuracy and precision of the instrument was determined on $n$-alkanoic acid ester standards of known isotopic composition (Indiana standard F8–3). Each sample was measured in replicate (with mean of S.D. 0.11‰ for $C_{16:0}$ and 0.10‰ for $C_{18:0}$). Values were also corrected to account for the methylation of the carboxyl group occurring during acid extraction.

**Biomolecular contamination control.** A number of measures were adhered to minimize the effects of contamination from modern proteins. Samples of calcite and pottery matrix were sampled by first removing approximately 1mm of the outer surface. We also took controls of the outside of the pottery matrix in order to identify any proteins which may be present owing from the soil, or derive from sample washing or handling. Prior to extraction, an initial wash step was performed through the addition of nuclease free EDTA, which was rotated for 5 min. The supernatant from this wash was subsequently removed and not analyzed and the residual sample was then subject to protein extraction. Extraction was performed in a dedicated ancient proteomics clean room. We also included three blank laboratory extractions with our analysis, as well as instrument washes between each MS/MS run. Polyethylene glycol (PEG) was also observed, which may have derived from the storage of samples in polythene bags. Despite these precautions, we observed a number of human-derived proteins in some of the samples, including keratins, uromodulin and dermcidin. We suspect that some of these human-derived proteins may have been introduced at the time of excavation and sample handling, as the majority are derived from epithelial cells. In the extraction blanks sent to both MS/MS locations, we observed keratin proteins. In the instrument washes from one set of LC-MS/MS analyses, we observed human blood proteins which may represent carry-over. In some samples, we saw an identified protein which may be sourced from a foodstuff, in addition to human homologue protein. For example, in CW18, serum albumin from both *Homo sapiens* and *Capra hircus* was identified, with peptides matching uniquely to both taxa. In these cases, we have excluded the *Capra hircus* identification in the event it may be a misassignment to a human-derived protein. While these may indeed represent endogenous proteins from food sources, we err on the side of caution and exclude them as not to generate a false-positive identification of a food product. Measures to minimize contamination from exogenous lipids include sterilization of glassware in a furnace at a temperature of 450 °C for 6 h and extraction blanks.

**ZooMS.** Samples of animal bone from three approximately contemporaneous[25] areas of the West Mound (Trenches 1, 5, 7) were analyzed using collagen peptide mass fingerprinting, using previously published protocols and analysis[72] (Supplementary Data 4). Of 120 total samples, 99 displayed a peptide marker diagnostic of sheep collagen and 16 of goat. Five samples produced insufficient collagen for taxonomic determination.

**Mixing model.** To investigate the impact of mixing of different foodstuffs on $\Delta^{13}C$ values, the $\delta^{13}C_{16:0}$ values of ruminant adipose and dairy were defined using the published ranges obtained from modern African ruminants with access to both $C_3$ and $C_4$ sources (i.e., −35 to −15‰). The range of $C_3$ plants available were

conservatively estimated to be between −35 and −30‰ based on previous analysis of modern foodstuffs[4,73]. Regression equations between $\delta^{13}C_{16:0}$ and $\delta^{13}C_{18:0}$ were then obtained from published modern reference fat values[4,47,73–78] for modern dairy, ruminant adipose and plants (Supplementary Table 3). Possible $\delta^{13}C_{16:0}$ values for plant, ruminant adipose and dairy fats were randomly selected ($n = 100,000$). Next corresponding values for $\delta^{13}C_{18:0}$ were determined using a random process that takes into account uncertainties in regression slope and intercept. Concentration values ($n = 100,000$) for $C_{16:0}$ and $C_{18:0}$ in each food product were also selected using from the data obtained from the USDA database[79]. Finally, $\Delta^{13}C$ ($\delta^{13}C_{18:0} − \delta^{13}C_{16:0}$) values were calculated by drawing (100,000 iterations) from the randomized isotope values (above), accounting for the amount of fatty acid in each foodstuff.

## Data availability

Raw and processed MS/MS data from blanks, instrument washes and samples are available to download via the PRIDE partner repository under accession code PXD008647 and DOI: 0.6019/PXD008647. The authors declare that all other data supporting the findings of this study are available within the paper and its supplementary information files.

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

## Acknowledgements

We thank the Mass Spectrometry Laboratories at the Target Discovery Institute, University of Oxford and the Novo Nordisk Foundation Centre for Protein Research at the University of Copenhagen for mass spectrometry analysis. We thank Adrian Whitwood for XRF and XRD analysis. We acknowledge the support of the German Research Foundation (DFG), the Max Planck Society, a British Academy/Leverhulme Trust Small Research Grant (SG132859), the Danmarks Grundforskinngfond Niels Bohr Professorship DNRF128, and the Arts and Humanities Research Council (AH/L00691X/1) for financial support of this study. Work at Novo Nordisk Foundation Center for Protein Research (CPR) is funded in part by a generous donation from the Novo Nordisk Foundation (grant number NNF14CC0001). We thank the Çatalhöyük Research Project, namely Ian Hodder, for organizational support. We thank the Ministry of Culture and Tourism of the Republic of Turkey. We thank Amy Bogaard, Ester Oras, Christina Warinner and Ulf Schoop for helpful comments on a previous version of the manuscript. Thanks are also due to Alisa Hujić, Alisa Scheibner and Carolin Jauß for the discussions that sparked the idea for this study.

## Author contributions

E.R., J.H. and O.C. conceived of the project. A.C.C, J.H., R.F., L.S., R.J.-C. and M.M. performed the experimental work and GC-MS, GC-C-IRMS and LC-MS/MS analyses. R.F., A.C.C. and O.C. performed the mixed lipid data analysis and interpretation. I.F. assessed pottery characteristics and provided photographs. I.F., D.O., J.A. and E.R. selected the samples. J.A. provided insights into excavation contexts. E.S. provided archaeobotanical insight. D.O. provided archaeozoological insight and ZooMS data. C.S., O.C., E.R., J.O., N.B. and M.J.C. provided the resources and data interpretation. P.F.B. and E.R. co-directed West Mound excavations. E.R., J.H. and O.C. wrote the manuscript, with contribution from all authors.

## Additional information

**Competing interests:** The authors declare no competing interests.

