## [Peer Review File · Nature Communications]

Reviewers' comments:

Reviewer #1 (Remarks to the Author):

The content of the paper by Jessica Hendy et al "Ancient proteins from ceramic vessels at Çatalhöyük West reveal the hidden cuisine of early farmers" clearly demonstrates the claim of the title.

The paper is particularly well written, uses apart bottom-up proteomics, lipidomics and isotopic ratio mass spectrometry forming a set of evidence in agreement with the archeological data from bones and seeds found in the site. The proteomics analysis has been performed by two independent laboratories which is a proof of the care put in obtaining the experimental results.

As described in the introduction it is the first time that bottom-up proteomics is used to prove ancient cooking and to decipher the ingredients used at the Neolithic period. The paper may be to be published in Nature communications after a major revision answering the comments below.

1) At a first glance the samples are heavily contaminated by PEGs. As a consequence the Mascot are rather low for analysis done on an Orbitrap Lumos or Q-Exactive. The Referee performed the interrogations on the data in the PRIDE repository using PEAKS 7 which allow de novo sequencing. De novo sequencing is required as the genome of the dairy and non-dairy animals at the Neolithic period are not known. Generally PEAKS 7 identifies also much more proteins compared to the Mascot identification. Apart from the above mentioned PEG in nearly all samples several human contaminant proteins were found with a high number of unique peptides and appear as the major proteins. These samples are much more contaminated than any archeological sample analyzed in the Referee's laboratory. The two contaminations should be mentioned as the authors stated that they took great care to avoid contamination. Excel sheets with all the identified proteins must be given in the supplementary materials and the authors must explained how they have filtered the data to obtain the information they present. After reprocessing about half of the data the Referee retrieved only partially the information given by the authors. The processing information are as much important as the raw data.

2) In Table S2. Protein and Peptide Identifications the protein accession number must be given. The number of peptide spectrum matches (PSM) is NOT the most convincing argument. The number of unique peptide for each protein must be also given. And the unicity must be ascertained using BlastP. For what reason the Orbitrap Lumos was not operated at high resolution for MS/MS spectra? The data obtained on the Orbitrap Lumos operating at low MS/MS resolution as MS/MS spectra were acquired in the linear ion trap and on the Q-Exactive at high resolution must be compared and both results given in Table 2. When they are two Mascot scores for the same peptide in Peptides identification, are the peptides coming from different experiments?

3) The results from the first experiment and from the replication are in most of the cases contain a very different number of identified peptides (see for example CW23 Calcite Deposit and CW23 Calcite Deposit – Replication for which the number of identified peptides are respectively 126 and 6!). An explanation should be given for this discrepancy or the analysis performed a third time

Minor: The paragraph on Q-Exactive conditions at the end of MS/MS data analysis should be moved at the end of the MS/MS analysis.

4) Lipids has been analyzed using GC/MS after full hydrolysis. LC MS/MS on triacylglycerols (TAGs) affords much more precise information on the animal or vegetal species and may be performed using the mass spectrometers the authors used for proteomics. This kind of analyzes must be performed on selected samples. The odd number fatty acids are not discussed. They may be linear or cyclopropanated and the ratio of the two depends on milk processing. The nature of the odd number fatty acids must be determined and the information it brings discussed. As for the bottom-up proteomics data the lipidomics data should be given in a repository.

Reviewer #2 (Remarks to the Author):

The paper investigates protein residue from both animals and plant species, gained from potsherds. The ceramic fragments were analysed both taking carbon deposits on their surface and also, investigating the pottery matrix. This method has a wide range of forerunners in terms of lipid analysis and a weaker set of evidence in terms of botanical remains.

The source is a well-chosen, short-lived closed archaeological context within the excavation area of the West Mound at Çatal Höyük, dated between 5900 -5800 cal BC: some hundred years younger than the East Mound and thus with a stronger representation of the later phases of food producing lifeways in Anatolia. Archaeologically spoken, the samples are chosen with the highest care (stratification, articulated animal bones taken as control etc). The excavators of this site obviously worked closely together with the geochemists and other experts, in order to assure both the correct provenance and the optimal interpretation.

The paper follows a logical structure for presenting the wide range of the analyses, including XRD and XRF and 13 C stable isotope analyses, revealing the cooking of various legumes and cereals, fermented dairy products and also, the meat of caprinae and of bovides. The results with milk protein and its probable fermented form used for consumption are remarkable (reflecting on lactase intolerance). The primary merit of the paper is exactly this wide range of food processing, speaking for an intensive agriculture as compared with earlier Neolithic centuries in South West Asia.

With the methodology, there is a small technical remark (153-154), which suggests the use of animal products in post-firing treatments, alone by the fact that such remains were found on the external pottery surface. A more realistic explanation would be the simple pollution of the outer surface which often happens while cooking and pouring the meal. Another point is the unexplained involvement of zooarchaeological investigations of animal bones from other parts of the tell mound. It is not clear why this was necessary, once the faunal assemblages in Çatal have been already cleared. A more serious deficiency, in my eyes, is that although references are made to the analysis of the human dental calculus, such publications are cited, but there is no attempt to compare the food remains taken from

plaques on teeth and plaques on vessel fragments. This comparison could bring really exciting and ground-breaking results, especially if differences in age/sex/gender/social status could be detected in the individuals who consumed the meal cooked in the pots investigated in the present paper. As the text reveals (322), there were burials found in the very same deposits of abandoned buildings.

Last, a point of criticism goes to line 257-264. The intensification of agricultural practices is known (e.g. Colledge and Conolly 2016), and the statement about a same scenario that "might be true for the dispersal of pottery... or alternatively pottery technology may have hitchhiked..." seem a commonplace on the one hand, and a phenomenological degradation of the otherwise good results of the paper on the other.

E. Bánffy

Reviewer #3 (Remarks to the Author):

I think this is a very important, pioneering piece of work, showing how analysis of proteins associated with ancient pottery can inform on cooking and food practice. Its samples from the key site of Catalhoyuk give the study extra significance. The comparison with other sources of dietary information and especially with lipid analysis is revealing. I hope this work can be widely emulated elsewhere.

I have a few general comments for consideration.

Since the best results were from the calcified deposits on the pottery, I think it is important to give the reader just a little more detail on why those deposits could not possibly be post-depositional.

The first mention of the dairy story (lines 45-46) is too compressed. The fuller, more complicated and lengthier is set out well later in the paper, but the narrative is a little simplified at first introduction. I am sure there are more recent references than ref 50 to cite, in the plethora of recent aDNA papers.

If there is space, there is more that could be said at lines 262-3 about the character and use of pottery as it was diffused north-westwards into Europe.

Finally, a point about language for non-scientific experts like myself. If there is space, and what I suggest should take up little extra space, the general reader would benefit greatly from some plain-English definitions of eg lipids, proteins, tissues, taxonomy, target epitopes, etc,

One or two details:

line 78: first introduction of dating of Catal is slightly confusing. Sort by adding eg 'overall'.

line 139: cap C on Caprinae

Two occurrences of 'in situ', which does not need a hyphen (Latin: my kind of jargon!).

Reviewer #1 (Remarks to the Author)

The content of the paper by Jessica Hendy et al “Ancient proteins from ceramic vessels at Çatalhöyük West reveal the hidden cuisine of early farmers” clearly demonstrates the claim of the title.

The paper is particularly well written, uses a bottom-up proteomics, lipidomics and isotopic ratio mass spectrometry forming a set of evidence in agreement with the archeological data from bones and seeds found in the site. The proteomics analysis has been performed by two independent laboratories which is a proof of the care put in obtaining the experimental results.

As described in the introduction it is the first time that bottom-up proteomics is used to prove ancient cooking and to decipher the ingredients used at the Neolithic period. The paper may be published in Nature communications after a major revision answering the comments below.

We thank the review for his/her comments on the manuscript and believe the study has been improved as a result.

1) At a first glance the samples are heavily contaminated by PEGs. As a consequence the Mascot are rather low for analysis done on an Orbitrap Lumos or Q-Exactive. The Referee performed the interrogations on the data in the PRIDE repository using PEAKS 7 which allow de novo sequencing. De novo sequencing is required as the genome of the dairy and non-dairy animals at the Neolithic period are not known. Generally PEAKS 7 identifies also much more proteins compared to the Mascot identification.

We thank the reviewer for picking up on this PEG contamination and we agree that it is essential we mention this contamination. We have updated the manuscript text to include that PEG contamination was observed [Lines 470-472]. Although we are not completely certain where in the sampling or extraction pipeline PEG may have entered, we speculate that PEG contamination may have arisen from the storage of samples in polyethylene bags prior to subsampling.

While it is intriguing that PEAKS7 identifies more proteins compared to Mascot, we do not believe that performing de novo sequencing on these data is essential for this study. We think that there will not be substantial protein sequence variation between modern and ancient

domesticates, and hence a Mascot search against UniProt is sufficient for providing these identifications.

Apart from the above mentioned PEG in nearly all samples several human contaminant proteins were found with a high number of unique peptides and appear as the major proteins. These samples are much more contaminated than any archeological sample analyzed in the Referee's laboratory. The two contaminations should be mentioned as the authors stated that they took great care to avoid contamination.

We have created an additional supplementary table (Supplementary Table 3) to include all protein identifications made using Mascot, not only those related to putative foodstuffs. This includes the protein identifications found in extraction blanks. These were already provided as processed MZidentML files, but have been additionally provided in a new supplementary table (Supplementary Table 3). We observe contamination from human proteins, deriving from keratin and other skin proteins (which appears in the extraction blanks sent to both MS/MS locations), and from other human proteins, especially human serum albumin, which appears in the instrument washes and samples from one batch of LC-MS/MS analyses. We believe this contamination may have entered the system from residual peptides in the LC column during analyses. We have reported this in the manuscript [Lines 472 – 483] and have excluded all serum albumin identifications from our list of dietary identifications, even if they demonstrate specificity to non-human animal species. This has resulted in some minor alterations to Figure 3 and SI Figure 3.

Excel sheets with all the identified proteins must be given in the supplementary materials and the authors must explain how they have filtered the data to obtain the information they present. After reprocessing about half of the data the Referee retrieved only partially the information given by the authors. The processing information is as much important as the raw data.

We have created an additional supplementary table (Supplementary Table 3) which contains all protein identifications made using Mascot, not only those related to putative foodstuffs. Supplementary Table 3 contains filtered outputs of Mascot. We have retained Supplementary Table 2 as a list of only food-related protein and peptide identifications for ease of viewing

which dietary-related proteins were identified. We have also updated the methods section to include further details about our data filtering process [Lines 416 – 419].

2) In Table S2. Protein and Peptide Identifications the protein accession number must be given.

We have updated Supplementary Table 2 to include the UniProt protein accession number.

The number of peptide spectrum matches (PSM) is NOT the most convincing argument. The number of unique peptide for each protein must be also given.

We have updated Supplementary Table 2 to include the number of unique peptides. Each observed peptide spectrum match can still be viewed in the Peptide Identifications tab of Supplementary Table 2.

And the unicity must be ascertained using BlastP.

All identified dietary peptides were searched against BlastP. This is listed in the “Peptide Identifications” tab of Supplementary Table 2. Their unicity is listed in the column “Blast taxonomic hit”.

For what reason the Orbitrap Lumos was not operated at high resolution for MS/MS spectra?

The data obtained on the Orbitrap Lumos operating at low MS/MS resolution as MS/MS spectra were acquired in the linear ion trap and on the Q-Exactive at high resolution must be compared and both results given in Table 2.

We performed two sets of LC-MS/MS analyses to give complementary data (HCD vs CID fragmentation, higher sensitivity for MS/MS scans in the ion trap as compared to the Orbitrap), rather than just pure replications. SI Table 2 displays which peptides were identified using both the Lumos and Q-Exactive. We have now also mentioned in the manuscript that these two sets of analyses are complementary, rather than simply replications [Line 352, Line 388].

When they are two Mascot scores for the same peptide in Peptides identification, are the peptides coming from different experiments?

We believe this may have been an error caused by Excel formatting. Each line represents one peptide-spectral match so this table should only contain one Mascot score per peptide. It is possible that the decimal point was listed a comma, as is the practice for German number formatting. These settings have been updated, and now there is only one Mascots score per peptide identification.

3) The results from the first experiment and from the replication are in most of the cases contain a very different number of identified peptides (see for example CW23 Calcite Deposit and CW23 Calcite Deposit – Replication for which the number of identified peptides are respectively 126 and 6!). An explanation should be given for this discrepancy or the analysis performed a third time.

It is indeed the case that the first and second LC-MS/MS analyses yielded different numbers of identified proteins. This is because of the differing instrumentation (i.e. the first analysis on the Fusion Lumos, and the second analyses on an Orbitrap Q-Exactive). The Fusion Lumos higher sensitivity due to its larger ion transfer tube profile (about x2 for peptides). In addition the MS/MS detection in the ion trap detector (Lumos) is more sensitive than the detection in the Orbitrap (Q-Exactive) which elevates a larger number of peptides above the thresholds for detection and sufficient MS/MS spectral quality to meet FDR estimation criteria. We have written that these two sets of LC-MS/MS analysis are on different instrumentation and that this is a complementary analysis, rather than a simple replication [Lines 352 and 388]. We have updated SI Table 2 to make it clearer which identifications were made using the Lumos and which were made using the Q-Exactive, enabling comparison of the data in Table 2.

Minor: The paragraph on Q-Exactive conditions at the end of MS/MS data analysis should be moved at the end of the MS/MS analysis.

This paragraph has been moved to the end of the MS/MS analysis section.

4) Lipids has been analyzed using GC/MS after full hydrolysis. LC MS/MS on triacylglycerols (TAGs) affords much more precise information on the animal or vegetal species and may be performed using the mass spectrometers the authors used for proteomics. This kind of analyzes must be performed on selected samples. The odd number fatty acids are not discussed. They may

be linear or cyclopropanated and the ratio of the two depends on milk processing. The nature of the odd number fatty acids must be determined and the information it brings discussed. As for the bottom-up proteomics data the lipidomics data should be given in a repository.

We performed HT-GC on a selected number of samples with the best lipid preservation but found that TAGs were not preserved. We have clarified this in the main text of the manuscript, which now reads “Calcified and ceramic samples were also extracted using established methods to determine the presence of acyl lipids (mono-, di- and triglycerides)^{44,45} but extensive degradation has prevented their survival in the samples.” [Line 184-187]

We do not believe that the fatty acid ratios are very useful for tracing the contents of the pottery as these relative abundances are altered in the post-depositional environment. The odd-chain (branched) fatty acids (C17, C15) are sometimes used as indicator of dairy or ruminant products but also may be derived from bacteria present in the soil. We have commented on the presence of these in the text [Line 182]. We have also now reported the full range of fatty acids encountered in the SI Table 8 including the range of branched chain fatty acids.

There is no online repository for lipidomics data from archaeological contexts. The degraded nature of the lipids encountered makes application of the lipidomics approach (with comparison with modern reference fats) problematic. We have however provided a table with the full range of lipids identified in each of the samples (SI Table 8).

Our interpretations are largely based on inferring the isotope values as in previous studies used to investigate dairying in the archaeological record. This approach is well established and justified as we know, from experimental studies, that these values are not affected by degradation in the burial environment.

Reviewer #2 (Remarks to the Author):

The paper investigates protein residue from both animals and plant species, gained from potsherds. The ceramic fragments were analysed both taking carbon deposits on their surface and also, investigating the pottery matrix. This method has a wide range of forerunners in terms of lipid analysis and a weaker set of evidence in terms of botanical remains.

The source is a well-chosen, short-lived closed archaeological context within the excavation area of the West Mound at Çatal Höyük, dated between 5900 -5800 cal BC: some hundred years younger than the East Mound and thus with a stronger representation of the later phases of food producing lifeways in Anatolia. Archaeologically spoken, the samples are chosen with the highest care (stratification, articulated animal bones taken as control etc). The excavators of this site obviously worked closely together with the geochemists and other experts, in order to assure both the correct provenance and the optimal interpretation.

The paper follows a logical structure for presenting the wide range of the analyses, including XRD and XRF and ¹³C stable isotope analyses, revealing the cooking of various legumes and cereals, fermented dairy products and also, the meat of caprinae and of bovides. The results with milk protein and its probable fermented form used for consumption are remarkable (reflecting on lactase intolerance). The primary merit of the paper is exactly this wide range of food processing, speaking for an intensive agriculture as compared with earlier Neolithic centuries in South West Asia.

We thank the reviewer for her comments on the manuscript.

With the methodology, there is a small technical remark (153-154), which suggests the use of animal products in post-firing treatments, alone by the fact that such remains were found on the external pottery surface. A more realistic explanation would be the simple pollution of the outer surface which often happens while cooking and pouring the meal.

This could indeed be a possibility, and we have added a sentence to note this [Line 159].

Another point is the unexplained involvement of zooarchaeological investigations of animal bones from other parts of the tell mound. It is not clear why this was necessary, once the faunal assemblages in Çatal have been already cleared.

The only zooarchaeological data included is novel data produced using Zooarchaeology by Mass Spectrometry, with the purpose of determining the proportion of sheep and goat remains at the site, in order to supplement the species identifications made using the identification of milk proteins. This ZooMS data has not been published previously elsewhere.

A more serious deficiency, in my eyes, is that although references are made to the analysis of the human dental calculus, such publications are cited, but there is no attempt to compare the food remains taken from plaques on teeth and plaques on vessel fragments. This comparison could bring really exciting and ground-breaking results, especially if differences in age/sex/gender/social status could be detected in the individuals who consumed the meal cooked in the pots investigated in the present paper. As the text reveals (322), there were burials found in the very same deposits of abandoned buildings.

The burials found in the same deposits are two neonates, and therefore an analysis of dental calculus cannot be performed. We have added clarification that these burials are neonates [Line 327]. Comparing food remains from dental calculus and ceramics at Catalhoyuk would indeed be a very interesting study. However, the analysis in this study is focused on Catalhoyuk West, where there are no adult burials. At Catalhoyuk East, where there are several hundred burials, there is no data on dental calculus analysis currently published or available. We do agree that this would be a very exciting and ground-breaking future study, allowing a link to be made between food processing and consumption, but this is beyond the reach of this current work as the data/material is not accessible.

Last, a point of criticism goes to line 257-264. The intensification of agricultural practices is known (e.g. Colledge and Conolly 2016), and the statement about a same scenario that “might be true for the dispersal of pottery... or alternatively pottery technology may have hitchhiked...” seem a commonplace on the one hand, and a phenomenological degradation of the otherwise good results of the paper on the other.

We have revised and simplified our argument in this section. Here we argue that, based on the observation of plant foods and mixed foodstuffs, it is possible that Neolithic pottery vessels may have had a much wider function (i.e. beyond meat and milk) than previously thought. This now reads;

“It is conceivable that other pottery vessels from early farming sites in Anatolia, the Near East, Europe and Africa, also had a much wider range of uses, beyond the processing of milk and meat as often suggested by their predominance in lipid analysis^{1,46,48,50}. If so, the invention of pottery in the Near East at the end of the 8th millennium BC⁵¹ and its subsequent diffusion with the

expansion of the Neolithic after the mid 7th millennium⁵², which at Çatalhöyük corresponds to the intensification of agricultural practices^{32,53}, may have also been driven by the need to process agricultural produce and rather than simply animal products alone.” [Lines 262-268].

E. Bánffy

Reviewer #3 (Remarks to the Author):

I think this is a very important, pioneering piece of work, showing how analysis of proteins associated with ancient pottery can inform on cooking and food practice. Its samples from the key site of Catalhoyuk give the study extra significance. The comparison with other sources of dietary information and especially with lipid analysis is revealing. I hope this work can be widely emulated elsewhere.

We thank the reviewer for his/her comments on the manuscript.

I have a few general comments for consideration.

Since the best results were from the calcified deposits on the pottery, I think it is important to give the reader just a little more detail on why those deposits could not possibly be post-depositional.

We have expanded our justification as to why we believe these deposits are post depositional, i.e. that we find little depositional build on the ceramic breaks, that we see the vast majority of the deposit on the internal surface of the ceramic. We have added that no post-depositional build-up on bone material excavated from the same context was recorded [Lines 101-104].

The first mention of the dairy story (lines 45-46) is too compressed. The fuller, more complicated and lengthier is set out well later in the paper, but the narrative is a little simplified at first introduction. I am sure there are more recent references than ref 50 to cite, in the plethora of recent aDNA papers.

We have expanded this section to further introduce the topic in line with the narrative in the discussion. We also include more citations to ancient DNA studies. This now reads;

“In Southwestern Asia, Africa and Europe, lipid residue analysis has fundamentally advanced our understanding of the development of early pastoral economies. Together with ancient DNA analysis, this has shown that early, and most likely lactose intolerant, farmers consumed dairy products by the time that the appearance of pottery first allows for the detection of this foodstuff^{1,6-8}. Although this research has far reaching consequences for understanding the emergence of dairying and the subsequent evolution of cultural, dietary and economic practices, lipids often lack taxonomic and tissue specificity (i.e. which species and/or which part of the animal/plant was consumed).” [Lines 47 – 54].

If there is space, there is more that could be said at lines 262-3 about the character and use of pottery as it was diffused north-westwards into Europe.

Upon request of Reviewer 2 we have revised and simplified our argument here. With this dataset, we can only speculate that Neolithic pottery vessels may have had a much wider function (i.e. beyond meat and milk) than previously thought. This section now reads;

“It is conceivable that other pottery vessels from early farming sites in Anatolia, the Near East, Europe and Africa, also had a much wider range of uses, beyond the processing of milk and meat as often suggested by their predominance in lipid analysis^{1,46,48,50}. If so, the invention of pottery in the Near East at the end of the 8th millennium BC⁵¹ and its subsequent diffusion with the expansion of the Neolithic after the mid 7th millennium⁵², which at Çatalhöyük corresponds to the intensification of agricultural practices^{32,53}, may have also been driven by the need to process agricultural produce and rather than simply animal products alone.” [Lines 262-268].

Finally, a point about language for non-scientific experts like myself. If there is space, and what I suggest should take up little extra space, the general reader would benefit greatly from some plain-English definitions of eg lipids, proteins, tissues, taxonomy, target epitopes, etc,

We have provided non-technical explanations of some of these terms where they are first mentioned in the text. E.g. “lipids often lack taxonomic and tissue specificity (i.e. which species and which part of the animal/plant was consumed).” [Line 54]; “specific target epitopes (regions of a protein which enable antigen-antibody binding)” [Line 69].

One or two details:

line 78: first introduction of dating of Catal is slightly confusing. Sort by adding eg 'overall'.

We have updated this sentence to clarify the dating phases. This now reads: “Here we applied a shotgun proteomic approach to ceramic sherds from the early farming site of Çatalhöyük (ca. 7,100 - 5,600 cal BC 23–25) in central Anatolia. Specifically, sherds were analyzed from The West Mound (Figure 1), radiocarbon dated to 6,000 - 5,600 cal BC[25]. The West Mound represents a stage within a process of socio-economic change (beginning on the East Mound at ca. 6,500 cal BC)...” [Lines 83 – 86].

line 139: cap C on Caprinae

This has been updated.

Two occurrences of 'in situ', which does not need a hyphen (Latin: my kind of jargon!).

These two instances have been updated.

REVIEWERS' COMMENTS:

Reviewer #1 (Remarks to the Author):

As the authors pointed out the quality of the analytical part has been much improved and after a careful reading of the modified text and tables I noted that happily all the remarks or questions on the analyses have been addressed conclusively.

Now this version of the article deserves to be published in Nature.

A comment: I do not agree with the sentence "While it is intriguing that PEAKS7 identifies more proteins compared to Mascot, we do not believe that performing de novo sequencing on these data is essential for this study. We think that there will not be substantial protein sequence variation between modern and ancient domesticates, and hence a Mascot search against UniProt is sufficient for providing these identifications." Even in different modern races of cattle's they are differences which can be found using de novo search. But indeed Mascot and BlastP search are enough for obtaining the desired answer.

Christian Rolando
PhD, CNRS senior scientist